# Impact of Hypermannosylation on the Structure and Functionality of the ER and the Golgi Complex

**DOI:** 10.3390/biomedicines11010146

**Published:** 2023-01-06

**Authors:** Patricia Franzka, Svenja Caren Schüler, Takfarinas Kentache, Robert Storm, Andrea Bock, Istvan Katona, Joachim Weis, Katrin Buder, Christoph Kaether, Christian A. Hübner

**Affiliations:** 1Institute of Human Genetics, University Hospital Jena, 07747 Jena, Germany; 2Leibniz-Institute on Aging—Fritz-Lipmann-Institute, 07745 Jena, Germany; 3De Duve Institute, UCLouvain, BE-1200 Woluwe-Saint-Lambert, Belgium; 4Molecular Devices, 81377 München, Germany; 5Institute of Neuropathology, RWTH Aachen University Hospital, 52074 Aachen, Germany

**Keywords:** Golgi network, endoplasmic reticulum, mannosylation

## Abstract

Proteins of the secretory pathway undergo glycosylation in the endoplasmic reticulum (ER) and the Golgi apparatus. Altered protein glycosylation can manifest in serious, sometimes fatal malfunctions. We recently showed that mutations in GDP-mannose pyrophosphorylase A (GMPPA) can cause a syndrome characterized by alacrima, achalasia, mental retardation, and myopathic alterations (AAMR syndrome). GMPPA acts as a feedback inhibitor of GDP-mannose pyrophosphorylase B (GMPPB), which provides GDP-mannose as a substrate for protein glycosylation. Loss of GMPPA thus enhances the incorporation of mannose into glycochains of various proteins, including α-dystroglycan (α-DG), a protein that links the extracellular matrix with the cytoskeleton. Here, we further characterized the consequences of loss of GMPPA for the secretory pathway. This includes a fragmentation of the Golgi apparatus, which comes along with a regulation of the abundance of several ER- and Golgi-resident proteins. We further show that the activity of the Golgi-associated endoprotease furin is reduced. Moreover, the fraction of α-DG, which is retained in the ER, is increased. Notably, WT cells cultured at a high mannose concentration display similar changes with increased retention of α-DG, altered structure of the Golgi apparatus, and a decrease in furin activity. In summary, our data underline the importance of a balanced mannose homeostasis for the secretory pathway.

## 1. Introduction

Glycosylation is the most common post-translational modification of proteins and lipids. It is relevant for the majority of plasma membranes and secreted proteins affecting their stability and conformation [1,2,3]. Thus, glycosylation plays a prominent role in many biological processes including cell-to-cell communication, cell-matrix interaction, adhesion, protein targeting and folding, viral or bacterial infection, cancer, and aging [1,3,4].

Glycosylation starts in the cytoplasm with the generation of glycosyl donors, the activated form of sugars or sugar derivatives. This activation requires specific enzymes, which produce nucleotide-diphosphate-sugars using nucleotide-triphosphate as a substrate [5,6]. Activated sugars are linked to the cytoplasmic site of dolichol-phosphate at the endoplasmic reticulum (ER). Then, dolichol-phosphate-sugars are flipped into the ER and the carbohydrate chain is transferred onto nascent protein chains [7]. In the ER, carbohydrates are trimmed sequentially by ER-resident glycosidases coordinated by folding and quality control factors. In the Golgi apparatus both glycan trimming and branch extension occur for generating mature glycans [8,9].

Abnormal glycosylation of proteins can induce ER- and Golgi-stress and can finally manifest in serious, sometimes fatal malfunctions of different organ systems such as the brain and muscle [10]. GMPPB-associated neuromuscular disorders are typical examples. GMPPB encodes the enzyme GDP-mannose-pyrophosphorylase-B (GMPPB), which is required to provide GDP-mannose (activated mannose) as a sugar donor for glycosylation [11]. We recently reported that mutations of its catalytically inactive homolog GDP-mannose-pyrophosphorylase-A (GMPPA) cause AAMR syndrome, a disorder characterized by achalasia, alacrima, mental retardation, and muscle weakness [12]. Homozygous GMPPA KO mice recapitulate many features of the human AAMR syndrome, including cognitive impairment and progressive muscle weakness. KO mice also show structural brain alterations and progressive neurodegeneration. We and others provided evidence that GMPPA acts as an allosteric feedback inhibitor of GMPPB [13,14]. Thus, its disruption elevates cytoplasmic GDP-mannose levels, which increases the incorporation of mannose into glycochains of various proteins [13].

Mutations in proteins necessary for the structure of the ER and the Golgi apparatus can also lead to the aberrant glycosylation of proteins. Prominent examples are mutations in components of the conserved oligomeric complex (COG), which is a hetero-octameric peripheral membrane protein complex required for retrograde vesicular transport and glycoconjugate biosynthesis within the Golgi apparatus [15]. Mutations in subunits 1, 4, 5, 6, 7, and 8 of this complex can cause different congenital disorders of glycosylation [16]. Defects of proteins involved in the trafficking between ER and Golgi apparatus such as syntaxin-5 can also affect the structure of the ER and the Golgi apparatus and result in changes of protein glycosylation [17]. Here, we assessed different tissues and cell types of GMPPA KO mice to study the possible consequences of enhanced mannosylation on the secretory pathway.

## 2. Materials and Methods

All animal experiments were approved by the Thüringer Landesamt für Lebensmittelsicherheit und Verbraucherschutz (TLLV) in Germany (approval number UKJ-18-024 and UKJ-17-006). Experiments were performed on WT and GMPPA KO litter mates on a C57BL/6 background in the 4th generation. GMPPA KO mice were generated as described previously [13]. Mice were housed in a 12 h light/dark cycle and fed on a regular diet ad libitum. Only male mice were used for tissue analysis. Experiments were performed at different ages as indicated.

### 2.1. Histological Analysis of Murine Tissue

Mice were sacrificed and organs removed. Muscles (Musculus Tibialis anterior) were fixed in 4% paraformaldehyde (PFA) and washed with 1× phosphate buffered saline (PBS). After incubation in PBS with 0.25% Triton-X-100 (Merck, Darmstadt, Germany) overnight, fiber bundles were dissected. After blocking in 5% normal goat serum (NGS), fiber bundles were incubated with the following primary antibodies: rabbit anti-TGN38 1:250 (Santa Cruz, Heidelberg, Germany, sc-166594), rabbit anti-GLG1 1:250 (Abcam, Cambridge, United Kingdom, ab262704), and mouse anti-GM130 1:250 (BD Biosciences, Heidelberg, Germany, 610822) overnight. After washing with 1xPBS, fiber bundles were incubated with the corresponding secondary antibodies (Alexa-Fluor coupled antibodies, Thermo Fischer Scientific, Nidderau, Germany). Nuclei were stained with DAPI (Thermo Fischer Scientific, Nidderau, Germany, 10 µg/mL). Fiber bundles were washed with 1xPBS and mounted with Fluoromount-G (Southern Biotech, Biozol, Eching, Germany). Images were taken with a Zeiss LSM880 Airyscan confocal microscope (Zeiss, Jena, Germany) with the Z-stack module and a 63× objective. Images were further analyzed with ImageJ. Colocalization analysis was performed with the Coloc2 module in ImageJ. Manderson overlap coefficients were used to quantify colocalization. Particle numbers were quantified in ImageJ using binary pictures and the plugin for particle analysis. To facilitate and accelerate image analysis, the ComDet v.0.5.5 plugin in ImageJ was used for colocalization analysis and particle number measurement. Since both approaches showed similar results, the ComDet plugin was used for the following analyses.

Brains were cryo-sectioned into 8 µm thick sections. Immunofluorescence (IF) stainings were performed in Shandon chambers (Thermo Fischer Scientific, Nidderau, Germany). Sections were fixed with 4% PFA and rinsed in phosphate-buffered saline (PBS). An amount of 0.25% Triton-X in 1xPBS was used to permeabilize cells. Blocking with 5% NGS was followed by incubations with primary and secondary antibodies as indicated above and mounted with Fluoromount-G (Southern Biotech, Biozol, Eching, Germany). Images were taken with a Zeiss LSM880 Airyscan confocal microscope with the Z-stack module and a 63× objective. Images were further analyzed with ImageJ. Colocalization and particle numbers were analyzed using the ComDet v.0.5.5 plugin in ImageJ.

### 2.2. Mouse Embryonic Fibroblast (MEF) Experiments

MEFs were maintained in DMEM (Gibco, Thermo Fischer Scientific, Nidderau, Germany) and supplemented with 10% fetal bovine serum (FBS, Gibco, Thermo Fischer Scientific, Nidderau, Germany), penicillin (100 UI/mL), and streptomycin (100 mg/mL) (P/S, Gibco, Thermo Fischer Scientific, Nidderau, Germany).

For α-DG-GFP localization experiments, cells were plated on coverslips. Twenty-four hours later, cells were transfected with 1 µg plasmid DNA/well encoding α-DG-GFP with lipofectamine 2000 reagent (Thermo Fischer Scientific, Nidderau, Germany) according to the instructions of the manufacturer. Twenty-four hours post-transfection, cells were fixed with 4% PFA at room temperature, permeabilized with 0.25% Triton-X in 1xPBS, and blocked with 5% NGS. Immunofluorescence studies were performed as described above and sections were mounted with Fluoromount-G (Southern Biotech, Biozol, Eching, Germany). For IF stainings of untransfected cells, cells were plated on coverslips for 2 days and then fixed with 4% PFA, permeabilized with 0.25% Triton-X in 1xPBS, and blocked with 5% NGS, and immunocytochemistry stainings were performed as described above. Images were taken with a Zeiss LSM880 Airyscan confocal microscope with the Z-stack module and a 63× objective.

For IF experiments with supplementation of glucose or mannose, cells were plated in imaging plates (Ibidi, München, Germany). After 6 h, cells for α-DG-GFP localization experiments were transfected with 1 µg plasmid DNA/well encoding α-DG-GFP with lipofectamine reagent 2000 (Thermo Fischer Scientific, Nidderau, Germany) and medium was changed for all cells. Medium was supplemented with either 4.5 g/L glucose or with 4.5 g/L mannose. After 2 days, MEFs were fixed with 4% PFA at room temperature, permeabilized with 0.25% Triton-X in 1xPBS, and blocked with 5% NGS, and immunocytochemistry was performed as described above. The following primary antibodies were used: rabbit anti-TGN38 1:250 (Santa Cruz, Heidelberg, Germany, sc-166594), rabbit anti-GLG1 1:250 (Abcam, Cambridge, United Kingdom, ab262704), mouse anti-GM130 1:250 (BD Biosciences, Heidelberg, Germany, 610822), mouse anti-PDI 1:500 (Enzo, Lörrach, Germany, ADI-SPA-891), mouse anti-calnexin 1:500 (Merck, Darmstadt, Germany, MAB3126), sheep anti-CLIMP63 1:500 (R&D Systems, Wiesbaden, Germany, AF7355), and rabbit anti-RTN4 1:500 (Abcam, Cambridge, United Kingdom, ab47085). After washing, the appropriate secondary antibodies were used (Alexa-Fluor coupled antibodies, Thermo Fischer Scientific, Nidderau, Germany). After another washing round, cells were stained with DAPI (Thermo Fischer Scientific, Nidderau, Germany, 10 µg/mL), washed, and kept in 1XPBS (with calcium and magnesium) for image acquisition. Image acquisition was performed with a confocal high-content imaging microscope ImageXpress HT.ai (Molecular Devices, München, Germany) and a 40× objective. Images were further analyzed with ImageJ. Colocalization and particle numbers were analyzed using the ComDet v.0.5.5 plugin as well as the Coloc2 module in ImageJ (see Appendix A).

### 2.3. Western Blot

MEFs were cultured in DMEM (Gibco, Thermo Fischer Scientific, Nidderau, Germany) as indicated above. Cells were harvested with RIPA buffer. After, sonication homogenates were centrifuged at 16,900× *g* to remove nuclei and insoluble debris. The supernatant was stored at −80 °C. For protein concentration measurement, the Pierce BCA kit (Thermo Fischer Scientific, Nidderau, Germany) was used according to the manufacturer’s instructions. Proteins were denatured at 90 °C for 5 min in laemmli sample buffer. After separation via SDS-polyacrylamide gel electrophoresis, proteins were transferred onto PVDF membranes (Whatman, Dassel, Germany). Membranes were blocked in 2% bovine serum albumin (BSA, Merck, Darmstadt, Germany) in tris-buffered saline (TBS) buffer (50 mM Tris-Cl pH 7.6, 150 mM NaCl) supplemented with 0.1% Tween-20 (Merck, Darmstadt, Germany) for 1 h at room temperature. Membranes were incubated with primary antibodies at appropriate dilutions overnight at 4 °C: rabbit anti-GAPDH 1:1000 (Proteintech, Planegg-Martinsried, Germany, 10494-1-AP), rabbit GMPPA 1:500 (Proteintech, Planegg-Martinsried, Germany, 15517-1-AP), mouse anti-PDI 1:500 (Enzo, Lörrach, Germany, ADI-SPA-891), mouse anti-pan cadherin 1:500 (BD Biosciences, Heidelberg, Germany), sheep anti-dystroglycan 1:1000 (R&D Systems, Wiesbaden, Germany, AF6868), biotinylated lens culinaris lectin (LCH) 1:300 (Vectorlab, Biozol, Eching, Germany), and biotinylated concanavalin A lectin (Con A) 1:300 (Vectorlab, Biozol, Eching, Germany). Primary antibodies were detected with a horseradish peroxidase-conjugated secondary anti-rabbit antibody (Amersham Bioscience, Freiburg, Germany), anti-mouse antibody (Amersham Bioscience, Freiburg, Germany), anti-sheep antibody (Merck, Darmstadt, Germany), and Streptavidin antibody (Merck, Darmstadt, Germany), and the Super Signal Western Blot Enhancer Kit (BioRad, Feldkirchen, Germany). The quantification of bands was done with ImageJ. All blots were repeated at least once. Coomassie blue staining of PVDF membranes was performed after protein detection. PVDF membranes were fixed for 3 min (10% acetic acid, 40% EtOH), stained in Coomassie blue solution (0.1% Brilliant Blue R (Serva, Heidelberg, Germany), 45% EtOH (10% acetic acid) for 5 min, destained (10% acetic acid, 20–40% EtOH), rinsed in distilled H_2_O, and imaged [18].

### 2.4. Sugar Measurements

Supernatant was taken from confluent cells and frozen until further use. Sugar concentrations in the supernatant were measured with the D-mannose, D-fructose, D-glucose kit following the manufacturer’s instructions (Megazyme, Neogen Europe Ltd, Auchincruive, Scotland, K-MANGL).

### 2.5. Furin Activity Assay

Mice were sacrificed and brain tissue was immediately frozen in liquid nitrogen and stored at −80 °C until further use. Blood was taken from unfasted mice and incubated on ice for 15 min. Samples were centrifuged for 10 min at 4 °C and 4000× *g* and stored at −80 °C. Cultured MEFs were harvested, washed with 1XPBS, and centrifuged at 1000× *g* for 5 min. Cell pellets were stored at −80 °C until further use. For measuring furin activity, brain tissue and cells were homogenized in 1XPBS (supplemented with protease inhibitors) and centrifuged (Eppendorf, Wesseling, Germany) for 10 min at 4 °C and 10,000× *g*. Furin activity was measured in the supernatant according to the manufacturer´s instructions (SensoLyte^®®^ Rh110 Furin Activity, Anaspec, Eurogentech, Köln, Germany). Blood/serum samples were used directly.

### 2.6. GDP-Mannose Measurements

MEFs were cultured after supplementation of either 4.5 g/L glucose or 4.5 g/L mannose for 10, 30 min, 4 h, or 20 h. For sample preparation, media were removed, cells were washed with cold water, quickly frozen in liquid nitrogen, and 250 µL of methanol/chloroform (9:1) was added to the plate. Cells were scraped from culture plates and centrifuged at 16,900× *g* for 10 min. Supernatants were dried by speed vacuum, resuspended in 50 µL of methanol 50%, and analyzed by LC/MS, as described previously [13].

### 2.7. Glycoprotein Enrichment and Mass Spectrometry

For glycoprotein enrichment, 2 mg total protein from total brain or skeletal muscle lysates were incubated with concanavalin A (Con A) coupled agarose beads at 4 °C overnight, and washed with lysis buffer (20 mM Tris, 150 mM NaCl, 1% (*v*/*v*) TritonX-100, complete protease inhibitor, and complete phosphatase inhibitor (Roche, Berlin, Germany). Glycoproteins were eluted with 200 mM glycine buffer pH 2.5. After elution, 1 M Tris buffer pH 10.4 was added and samples were stored at −20 °C.

For proteomics analysis, samples were sonicated (Bioruptor Plus, Diagenode, Sart Tilman, Belgium) for 10 cycles (30 s ON/60 s OFF) at high setting, at 20 °C, followed by boiling at 95 °C for 5 min. Reduction was followed by alkylation with iodoacetamide (IAA, final concentration 15 mM) for 30 min at room temperature in the dark. Protein amounts were estimated following an SDS-PAGE gel of 10 µL of each sample against an in-house cell lysate of known quantity. An amount of 30 µg of each sample was taken along for digestion. Proteins were precipitated overnight at −20 °C after the addition of 8× volume of ice-cold acetone. The following day, the samples were centrifuged at 20,800× *g* for 30 min at 4 °C and the supernatant was carefully removed (Eppendorf 5810R, Eppendorf AG, Wesseling, Germany). Pellets were washed twice with 300 µL ice-cold 80% (*v*/*v*) acetone in water, then centrifuged at 20,800× *g* at 4 °C for 10 min. After removing the acetone, pellets were air-dried before addition of 25 µL of digestion buffer (1 M Guanidine, 100 mM HEPES, pH 8). Samples were resuspended with sonication as explained above; then, LysC (Wako, Braunschweig, Germany) was added at a 1:100 (*w*/*w*) enzyme:protein ratio and digestion proceeded for 4 h at 37 °C under shaking (1000 rpm for 1 h, then 650 rpm). Samples were then diluted 1:1 with MilliQ water, and trypsin (Promega, Walldorf, Germany) was added at a 1:100 (*w*/*w*) enzyme:protein ratio. Samples were further digested overnight at 37 °C under shaking (650 rpm). The day after, digests were acidified by the addition of TFA to a final concentration of 10% (*v*/*v*), heated at 37 °C, and then desalted with Waters Oasis^®®^ HLB µElution Plate 30 µm (Waters Corporation, Milford, MA, USA) under a soft vacuum following the manufacturer´s instruction. Briefly, the columns were conditioned with 3 × 100 µL solvent B (80% (*v*/*v*) acetonitrile; 0.05% (*v*/*v*) formic acid) and equilibrated with 3 × 100 µL solvent A (0.05% (*v*/*v*) formic acid in Milli-Q water). The samples were loaded, washed with 3 × 100 µL solvent A, and then eluted into 0.2 mL PCR tubes with 50 µL solvent B. The eluates were dried down using a speed vacuum centrifuge (Eppendorf Concentrator Plus, Eppendorf AG, Germany). Dried samples were stored at −20 °C until analysis.

For data acquisition, approximatively 1 μg of reconstituted proteins were separated using a nanoAcquity UPLC (Waters Corporation, Milford, MA, USA) that was coupled online to the MS. Peptide mixtures were separated in trap/elute mode, using a trapping (nanoAcquity Symmetry C18, 5 μm, 180 μm × 20 mm) and an analytical column (nanoAcquity BEH C18, 1.7 μm, 75 μm × 250 mm). The outlet of the analytical column was coupled directly to an Orbitrap Fusion Lumos mass spectrometer (Thermo Fisher Scientific, San Jose, California, USA) using the Proxeon nanospray source. Solvent A was water, 0.1% formic acid, and solvent B was acetonitrile, 0.1% formic acid. The samples were loaded with a constant flow of solvent A at 5 μL/min onto the trapping column. Trapping time was 6 min. Peptides were eluted via the analytical column with a constant flow of 300 nL/min. During the elution step, the percentage of solvent B increased in a nonlinear fashion from 0% to 40% in 90 min. Total runtime was 120 min, including cleanup and column re-equilibration. The peptides were introduced into the mass spectrometer via a Pico-Tip Emitter 360 µm OD × 20 µm ID; 10 µm tip (New Objective) and a spray voltage of 2.2 kV was applied. The capillary temperature was set at 300 °C. The RF lens was set to 30%. Full scan MS spectra with mass range 350–1650 *m*/*z* were acquired in profile mode in the Orbitrap with a resolution of 120,000 FWHM. The filling time was set at a maximum of 20 ms with an AGC target of 5 × 105 ions. DIA scans were acquired with 34 mass window segments of differing widths across the MS1 mass range. The HCD collision energy was set to 30%. MS/MS scan resolution in the Orbitrap was set to 30,000 FWHM with a fixed first mass of 200 *m*/*z* after accumulation of 1× 10^6^ ions or after filling time of 70ms (whichever occurred first). Data were acquired in profile mode. For data acquisition and processing, Tune version 2.1 and Xcalibur 4.1 were employed.

DIA data were then uploaded and searched against the spectral library in Spectronaut (v. 13, Biognosys AG). Relative quantification was performed in the software for each pairwise comparison using the replicates from each condition. The data (candidate table) and data reports were then exported as tables and further data analysis and visualization were performed with R-studio (version 0.99.902) using in-house pipelines and scripts. Data were then filtered for proteins that are known to be ER- and Golgi-resident and thus were assigned to the compartments “endoplasmic reticulum” and/or “Golgi”. These proteins are presented in Appendix A.

The mass spectrometry proteomics data have been deposited to the ProteomeXchange Consortium via the PRIDE [19] partner repository with the dataset identifier PXD014260 [13] and PXD037129.

### 2.8. Cell Fractionation of Transfected Cells

MEFs were cultured in DMEM (Gibco, Thermo Fischer Scientific, Nidderau, Germany) as indicated above. Cells were seeded and transfected with 1 µg plasmid DNA/well encoding α-DG-GFP in 10 cm dishes with lipofectamine 3000 reagent (Thermo Fischer Scientific, Nidderau, Germany) according to the instructions of the manufacturer. Medium was supplemented with either 4.5 g/L glucose or with 4.5 g/L mannose. After two days, cells were lysed in a homogenization buffer (0.25 M sucrose in 40 mM HEPES buffer pH 7.4, 120 mM KCl, supplemented with protease inhibitors (Roche, Berlin, Germany)) with an injection needle (21G) and a part was stored for total cell lysate fractions. The remaining lysates were centrifuged at 800× *g* for 10 min at 4 °C. Pellets (nuclei fraction) were lysed in tris-buffered saline (TBS) supplemented with 0.5% SDS and sonicated. Supernatants were supplemented with NP40, TritonX-100, and digitonin until reaching a final concentration of 1% and then passed through injection needles of different sizes (24G, 27G, 30G). Samples were centrifuged at 16,000× *g* at 4 °C for 1 h. Pellets (crude outer membrane fraction) were lysed in TBS + 0.5% SDS and sonicated. Supernatants were considered as cytoplasmic organelle fractions containing ER and Golgi. All fractions were supplemented with laemmli sample buffer and heated at 90 °C for 5 min. Then, samples were separated and immunoblotted as described above.

### 2.9. Electron Microscopy (EM) Analysis

Mice were perfused transcardially with 4% PFA and 2.5% glutaraldehyde in PBS. Skeletal muscle (gastrocnemius muscle) and the hippocampus were removed and postfixed. Afterwards, tissues were washed with 0.1 M cacodylate buffer pH 7.3 and postfixed in 1% osmium tetroxide/1% potassium ferrocyanide II in 0.1 M cacodylate buffer at 4 °C for 2 h. Tissues were dehydrated in the following accelerating acetone concentrations at 30%, 50%, 70%, 90%, 95%, and 100% (*v*/*v*). For contrast, 1% (*w*/*v*) uranyl acetate was added to 50% (*v*/*v*) acetone. The infiltration was carried out with epoxy resin. Processed biopsies were cut in 50 nm thick sections (Reichert Ultracut S, Leica, Wetzlar, Germany) and analyzed in a transmission electron microscope 900N (Zeiss, Jena, Germany) at 80 kV.

### 2.10. Statistical Analysis

For statistical analysis, raw data were analyzed for normal distribution with the Kolmogorov–Smirnov test or by graphical analysis using the Box-Plot and QQ-Plot in Graphpad prism 9. If appropriate, we either used 1-way ANOVA, 2-way ANOVA, or two-tailed Student’s t-tests. * indicates *p* < 0.05, ** *p* < 0.01, and *** *p* < 0.001. For statistical analysis, we used Graphpad prism 9. For all data, means with standard error of the mean (SEM) and individual data points with SEM are shown.

## 3. Results

### 3.1. The Structure of the Golgi Apparatus Is Altered upon Disruption of GMPPA

We used our recently established mouse model for AAMR syndrome [13] to study possible consequences of protein hypermannosylation for the ER and Golgi apparatus. To this end, we stained skeletal muscle fibers of WT and GMPPA KO mice for the cis-Golgi marker GM130 [20], the trans-Golgi network marker TGN38 [21], and the Golgi marker GLG1 [22]. We noted an increased number of GLG1- and TGN38-positive structures suggesting a fragmentation of the Golgi apparatus in skeletal muscle fibers of GMPPA KO mice. The mean particle size of GM130, TGN38, and GLG1 positive structures did not differ between genotypes, although we detected a trend towards a reduced particle size for TGN38 positive structures (data not shown). Moreover, the overlap between GM130 and TGN38 was diminished compared to WT mice (Figure 1A–H). For particle and colocalization analysis, we used different approaches: First, we assessed TGN38 and GM130 positively stained structures by converting the images to a binary format and then using the tool “analyze particles” in the ImageJ menu. Here, we detected significantly increased numbers of TGN38 and GLG1 positive structures. Secondly, we assessed particle numbers with the ImageJ plugin ComDet v.0.5.5 without prior image pre-processing. Here, we detected significantly increased GLG1 positive structures. For colocalization analyses, we used the ImageJ plugin Coloc2 that measures several colocalization coefficients, e.g., the Manderson overlap coefficient, and, detected a reduced colocalization between TGN38 and GM130 in KO samples. These latter results were confirmed with the ImageJ plugin ComDet v.0.5.5. Since we obtained similar data, we subsequently continued with the ComDet plugin.

Since AAMR patients and mice show cognitive and motor impairment [13], we wondered whether the Golgi apparatus is also affected in neurons. We analyzed Purkinje cells, which are progressively lost in GMPPA KO mice [13], because they have a particularly large soma, thus facilitating the analysis of the Golgi complex structure. To distinguish whether alterations of the Golgi apparatus in Purkinje cells are consequences of the neurodegenerative process or a direct effect of the loss of GMPPA, we assessed WT and GMPPA KO mice before onset of neuron loss in 3-month-old mice. Average particle size of GM130 and TGN38 positive structures was unchanged between genotypes (data not shown). The number of TGN38 positive structures was increased in Purkinje cells in GMPPA KO mice, while the colocalization between GM130 and TGN38 was reduced (Figure 1I,J,M,N), thus recapitulating our findings in skeletal muscle fibers.

We next assessed whether we could find similar alterations of the Golgi apparatus in mouse embryonic fibroblasts (MEFs) isolated from KO mice. In agreement with our in-vivo data, TGN38 positive particles and the overlap between GM130 and TGN38 positive compartments were decreased in KO MEFs (Figure 1K,L,O,P). Average particle size of GM130 and TGN38 positive structures did not differ between genotypes, although we detected a trend towards a reduced average particles size for TGN38 positive structures (data not shown).

Our findings were further strengthened by electron microscopy (EM) analysis of skeletal muscle and hippocampal neurons revealing enlarged secretory vesicles as well as enlarged Golgi cisternae, reduced Golgi stack numbers, and a fragmented trans-Golgi face (Figure 1Q–S).

Taken together, disruption of GMPPA results in structural alterations of the Golgi apparatus.

### 3.2. Proteomic Analysis of ER and Golgi Related Glycoproteins in Brain and Muscle Tissues of GMPPA KO Mice

Because of the prominent structural alterations of the Golgi apparatus, we wondered whether the abundance of ER- and Golgi-resident glycoproteins might be changed in GMPPA KO mice. To this end, we enriched mannosylated proteins from skeletal muscle and brain protein lysates from WT and KO mice by Concanavalin A (Con A) pulldown. The efficiency of the enrichment was validated by immunoblot analysis showing a strong increase in the Con A lectin signal in enriched samples (Appendix A). Subsequent mass spectrometry identified several ER- (e.g., Casq1) and some Golgi-resident (e.g., UGGG1) proteins, which were strongly regulated in either skeletal muscle or total brain lysates of KO mice (Appendix A). The Metascape analysis [23] suggested that mainly proteins involved in organelle stabilization, protein folding and trafficking, glycosylation, and protein phosphorylation were changed (Appendix A). Moreover, we generated violin plots for KO whole tissue lysates and Con A enriched samples showing all identified proteins in regard of their fold change compared to WT samples for the following processes: “Endoplasmic reticulum organization”, “Golgi organization”, “protease activity”, “protein maturation” and “protein processing”, and “vesicle mediated transport” (Figure 2A and Figure 3A). While full tissue lysates did not show a strong up- or down-regulation of proteins, Con A enriched proteins showed robust alterations (Figure 2A and Figure 3A). Because the abundance of most ER- and Golgi-resident proteins did not differ in total protein lysates (Appendix A, Figure 2A and Figure 3A), we conclude that differences detected upon Con A enrichment may reflect changes in the efficiency of proteins to be pulled down due to changes in their glycan chains or of their respective interaction partners. 

We also provide heatmaps for whole tissue lysates versus enriched samples for proteins found in the brain and skeletal muscle that were significantly altered between genotypes for the following processes: “Endoplasmic reticulum organization”, “Golgi organization”, “protease activity”, “protein maturation” and “protein processing”, and “vesicle mediated transport” (Figure 2B and Figure 3B). Notably, most of the proteins identified related to ER and Golgi organization and function are glycosylated. The proteins that are not glycosylated (Figure 2 and Figure 3) are known to interact with glycoproteins (Appendix A) and were thus enriched as well. Some of the glycoproteins identified upon Con A pulldown were not detected by MS of complete tissue lysates. These include the low abundant furin substrate Notch or the TGN sorting receptor sortilin in skeletal muscle and the ER calcium-storage protein calsequestrin in brain lysates. Of note, these proteins are mainly mannosylated [24,25,26].

In conclusion, Con A enrichment with subsequent mass spectrometry analysis shows that impaired glycosylation affects ER- and Golgi-resident proteins and a variety of cellular processes important for ER and Golgi structure, protein maturation, processing, and sorting. 

### 3.3. Golgi Function Is Altered in GMPPA KO Mice

Because of the prominent alteration of the Golgi complex structure upon deletion of GMPPA, we wondered about possible consequences for its function. Besides glycosylation, protein cleavage is a major function of the Golgi apparatus and the TGN. Although furin also cycles between TGN and the cell surface, this endoprotease predominantly localizes to the TGN. Furin has a wide range of substrates, including e.g., pro-β-nerve growth factor (pro-β-NGF), pro-bone morphogenetic protein-4 (pro-BMP-4), and the insulin pro-receptor [3,27]. We measured furin activity in murine blood as well as brain and MEF lysates and found that furin activity was decreased in GMPPA KO samples. This suggests that the function of the Golgi apparatus is changed in GMPPA KO mice (Figure 4A–C).

Because of the prominent alteration of the Golgi complex structure upon deletion of GMPPA, we wondered about possible consequences for its function. Besides glycosylation, protein cleavage is a major function of the Golgi apparatus and the TGN. Although furin also cycles between TGN and the cell surface, this endoprotease predominantly localizes to the TGN. Furin has a wide range of substrates, including e.g., pro-β-nerve growth factor (pro-β-NGF), pro-bone morphogenetic protein-4 (pro-BMP-4), and the insulin pro-receptor [3,27]. We measured furin activity in murine blood as well as brain and MEF lysates and found that furin activity was decreased in GMPPA KO samples. This suggests that the function of the Golgi apparatus is changed in GMPPA KO mice (Figure 4A–C).

### 3.4. Increased Retention of α-DG in the Absence of GMPPA

In our MS analysis, we identified several regulated ER-resident proteins in brain and skeletal muscle protein lysates of GMPPA KO mice, which raised the question of whether structural alterations of the Golgi apparatus in GMPPA KO mice come along with alterations of ER structure. To this end, we stained WT and GMPPA KO fibroblasts for phosphodiesterase (PDI), a luminal ER protein [28], and calnexin, a membrane bound ER resident protein that localizes mostly to ER sheets [29]. We did not detect differences in the distribution and intensity of PDI and calnexin signals between genotypes, neither in the number of PDI or calnexin positive structures nor in the colocalization coefficients between both proteins (Figure 4D,F,G).

To get an idea about the trafficking of mannosylated proteins from the ER to the Golgi apparatus and because GMPPA KO mice show a reduced abundance of α-DG in the sarcolemma of skeletal muscle fibers [13], we transfected WT and KO MEFs with a construct encoding α-DG-GFP. Co-staining with PDI allowed us to quantify the relative amount of α-DG-GFP within the ER. Notably, the overlap between α-DG-GFP and PDI was increased in KO fibroblasts, suggesting that the retention of α-DG-GFP in the ER is increased for KO MEFs (Figure 4E,H–J). Moreover, we fractionated cells into an organelle and crude outer membrane fraction after transfection with a dystroglycan construct. We used pan-cadherin as a marker for the plasma membrane and the luminal ER protein PDI as a marker for the organelle fraction. Notably, in WT cells, α- and β-DG were predominantly found in the crude membrane fraction, while in KO cells, both proteins were found in the organelle fraction (Appendix A).

### 3.5. GMPPA Associated Alterations of the ER and Golgi Apparatus Can Be Mimicked by Increasing the Extracellular Mannose Concentration

We recently showed that serum mannose concentrations are strongly increased in GMPPA KO mice [13], which is also evident in the supernatant of cultured MEFs isolated from GMPPA KO mice (Appendix A). Since we observed that dietary mannose restriction could at least in part prevent hypermannosylation and disease progression in GMPPA KO mice [13], we wondered whether mannose supplementation in WT MEFs might induce similar ER and Golgi alterations as observed in GMPPA KO mouse tissues or cells. Therefore, we cultured MEFs in the presence of either 4.5 g/L glucose or 4.5 g/L mannose. GDP-mannose levels drastically increased upon mannose supplementation in WT cells (Figure 5A). This resulted in increased protein mannosylation of WT cells as shown by Western blots of cell lysates probed with lens culinaris lectin (LCH) and Concanavalin A lectin (Con A) (Appendix A). Similar to our findings for GMPPA KO MEFs, we observed a reorganization of the Golgi apparatus in WT MEFs upon mannose treatment with increased numbers of GM130 and TGN38 positive structures, while the overlap between both markers was decreased. Notably, the effects of mannose supplementation were more prominent in WT compared to GMPPA KO cells (Figure 5B,C). Moreover, the colocalization of the predominant marker of ER tubules Reticulon-4 (RTN4) [30] and the predominant ER sheet marker CLIMP63 [31] was increased which may indicate a reorganization of the ER in response to increased mannose levels (Figure 5D,E).

We then overexpressed α-DG-GFP in MEFs and co-stained for the ER marker PDI or the Golgi marker GLG1. Notably, the overlap between α-DG-GFP with PDI was increased in MEFs cultured with mannose, while the overlap between α-DG-GFP with GLG1 was decreased. This suggests that the processing of α-DG-GFP in the ER is hampered by high mannose levels (Figure 6A–D). We also fractionated cells treated with either glucose or mannose into an organelle and crude outer membrane fraction after transfection with α-DG-GFP. We used pan-cadherin as a marker for the plasma membrane and the luminal ER marker PDI as a marker or the organelle fraction. Notably, in WT cells treated with glucose α- and β-DG were mostly found in the crude outer membrane fraction, while in KO cells as well as cells cultured with mannose both proteins were found in the organelle fraction (Appendix A).Supporting the notion that the function of the Golgi apparatus is altered in the presence of high mannose levels, furin activity was decreased in the supernatants of MEFs cultured with excessive mannose (Figure 6E). Taken together, these data indicate that extracellular mannose levels can impact the organization and function of the ER and the Golgi apparatus.

## 4. Discussion

Mutations in enzymes of the glycosylation machinery, which result in protein hypoglycosylation, can entail consequences for many cellular functions including protein processing in the ER and Golgi apparatus [32]. Whether the hyperglycosylation of proteins can also affect the function of the ER and the Golgi apparatus is so far elusive, which in part can be attributed to a lack of suitable model systems. We have recently shown that AAMR syndrome is caused by GMPPA loss of function. GMPPA regulates the activity of GMPPB, the enzyme that facilitates the production of GDP-mannose. Thus, its loss leads to increased GDP-mannose levels in the cytoplasm, which results in the increased incorporation of mannose into glycochains of different proteins including α-DG. Therefore, we used our GMPPA KO mice as a model organism to study the consequences of hypermannosylation for the Golgi apparatus. Our data show that protein hypermannosylation does not only result in alterations of the structure of the Golgi apparatus but also affects its functionality.

Usually, the Golgi apparatus is composed of a series of flat, parallel, interconnected cisternae organized around the microtubule-organizing center in the perinuclear region. The Golgi apparatus is a highly dynamic structure, which can reversibly disassemble during mitosis, when the Golgi complex forms clusters of vesicles that disperse throughout the cytoplasm [33,34]. Our morphological analysis suggests a fragmentation of the trans-Golgi network (TGN) in neurons and skeletal muscle fibers of GMPPA KO mice. Glycosyltransferases, glycosidases, and nucleotide sugar transporters are studded into Golgi membranes all over the Golgi complex [9]. Altered glycosylation of these proteins might affect their interaction with other proteins, their stability and turnover, and may directly affect the stability of the membrane. It has been reported that glycosylation may also influence the processing and functions of glycosyltransferases, including their secretion, stability, and the substrate/acceptor affinity [35]. Changes in these properties may have a profound impact on the activity of glycosyltransferases. For example, some glycosyltransferases have to be glycosylated themselves to become fully active. Consequently, altered glycosylation of glycosylation enzymes and possibly also nucleotide sugar transporters [36,37] might affect their trafficking along the Golgi apparatus, their function and integration into the Golgi membrane, and thus lead to membrane/cisterna disruption or enlargement. Moreover, proteins such as TMED2 and TMED10, which are highly mannosylated and control the formation of plasma membrane lipid nanodomains necessary for ER-Golgi membrane contact sites, are compromised in GMPPA KO mice. Many other glycoproteins that are affected in GMPPA KO mice, such as golgins (e.g., GOLGA2) [38], ankyrins [39,40], VAPA [41], SURF4, and ERGICs [42], are important for membrane trafficking and maintaining the structure of the Golgi complex. Thus, loss of GMPPA impacts various proteins necessary for trafficking and organization of ER and Golgi apparatus. Fragmentation of the Golgi apparatus and the TGN has been reported in several human neurodegenerative diseases, including Alzheimer’s disease (AD), amyotrophic lateral sclerosis (ALS), and Parkinson’s Disease [43,44].

To obtain further clues about the role of the fragmentation of the TGN in GMPPA KO mice, we enriched mannosylated proteins, which were then identified and quantified by mass spectrometry. Notably, several ER- and Golgi-resident proteins important for protein folding and trafficking, glycosylation, and protein phosphorylation are regulated in GMPPA KO samples. Whole tissue lysates did not show a strong up- or down-regulation of proteins, while Con A enriched proteins showed robust alterations. Thus, these alterations rather reflect changes in the efficiency of proteins to be pulled down due to changes in their glycan chains or their interaction partners.

Notably, the phosphofurin acidic cluster sorting protein1 (PACS1) was decreased in GMPPA KO tissues. PACS1, which is not glycosylated itself, plays a major role for the targeting of furin to the TGN [27,45] and may thus affect the activity of furin. Furin is a subtilisin-like proprotein convertase with a minimal cleavage site of Arg-X-X-Arg and preference for Arg-X-Lys/Arg-Arg. It is the major processing enzyme of the secretory pathway and localizes to the TGN. Substrates of Furin include blood clotting factors, serum proteins, growth factor receptors such as the insulin-like growth factor receptor, and α-DG. Furin is expressed in many tissues, including neuroendocrine tissues, the liver, gut, and brain. Furin undergoes an initial autocatalytic processing event in the ER and then locates to the trans-Golgi network through endosomes where a second autocatalytic event takes place and the catalytic activity is acquired [46,47]. Furin is mannosylated and sialylated [48]. Thus, loss of GMPPA might affect furin glycosylation.

Because of these findings, we measured furin activity in serum as well as brain and MEF lysates from WT and GMPPA KO mice. Notably, its activity was decreased in GMPPA KO samples. This can have important consequences for the processing of its substrates, including α-DG. Alpha-DG is a highly glycosylated extracellular peripheral membrane protein noncovalently attached to the transmembrane protein β-DG. Both proteins are encoded by a single gene (*DAG1*) and are generated by cleavage of a common precursor in the endoplasmic reticulum [49]. In the Golgi apparatus, α-DG is further processed by furin [50]. α-DG binds to ECM components such as laminin via its glycan side chains, and β-DG is connected to cytoplasmic proteins including dystrophin and thus acts as a linker between the ECM and the intracellular cytoskeleton, thereby stabilizing myofibers [51,52].

To assess the trafficking of a strongly mannosylated protein along the secretory pathway, we transfected WT and GMPPA KO MEFs with α-DG-GFP. Suggesting that the trafficking of α-DG is impaired, we found an increased retention of α-DG-GFP in the ER in the absence of GMPPA. Possibly, the altered structure of the Golgi apparatus impairs the trafficking of α-DG within the Golgi apparatus. Alternatively, the folding of α-DG-GFP may be impaired because of irregular glycosylation. In any case, this retention can partially explain the decreased abundance of α-DG in skeletal muscle fibers of AAMR patients and thus contribute to the pathogenesis of the myopathy associated with this disease. Since most glycoproteins are secreted and therefore undergo processing while trafficking through the ER and the Golgi apparatus [53], it is likely that similar effects also apply to other glycoproteins and non-glycosylated proteins. It will be of great interest to assess protein processing of endogenously tagged glycoproteins as well as non-glycosylated proteins in future studies.

We recently showed that serum mannose concentrations are strongly increased in GMPPA KO mice, which was also evident in the supernatant of cultured MEFs isolated from GMPPA KO mice. Increased mannose levels in the supernatant may reflect increased release from glycans and/or increased generation of mannose from glucose. Although only ~2% of mannose entering the cell is used for glycosylation [54,55], the higher systemic mannose levels may contribute to the larger pool of GDP-mannose in GMPPA KO mice and thus hypermannosylation.

Since we observed that dietary mannose restriction could at least in part prevent hypermannosylation and disease progression in GMPPA KO mice, we wondered whether mannose supplementation in WT MEFs could induce alterations as observed in GMPPA KO mice. Indeed, mannose induced a similar reorganization of the Golgi apparatus and the TGN in WT cells. Moreover, it decreased furin activity and promoted the retention of α-DG in the ER. It has been reported that external mannose is directly used for glycoprotein synthesis and our findings of elevated GDP-mannose levels in cells treated with mannose support this hypothesis [54,55,56]. We cannot exclude, however, that the effects of mannose supplementation may additionally interfere with other metabolic processes such as glycolysis, the tricarboxylic acid cycle, or the pentose phosphate pathway [57].

It is tempting to speculate that dietary mannose supplementation may also have detrimental effects in-vivo. To our knowledge, however, there are no reports addressing mannose intake on cell metabolism.

## Figures and Tables

**Figure 1 biomedicines-11-00146-f001:**
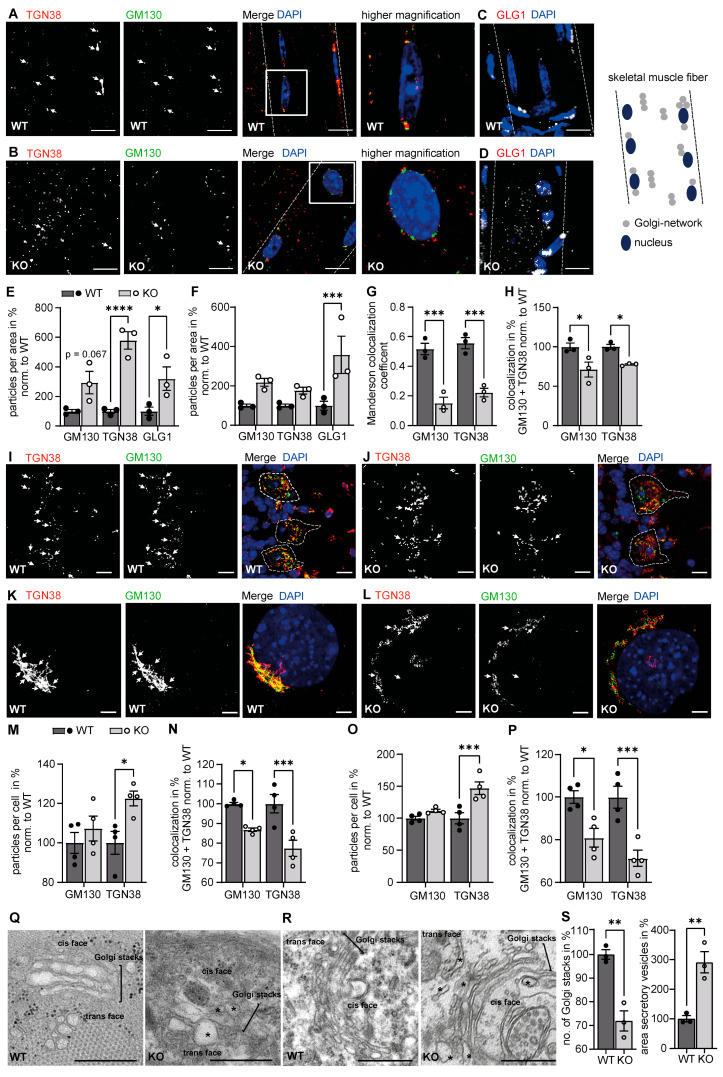
Loss of GMPPA alters the morphology of the Golgi apparatus in skeletal muscle fibers and in neuronal cells. (**A**,**B)** Representative images of skeletal muscle fibers from 5-month-old mice stained for TGN38 and GM130 (scale bar: 10 µm). (**A**) WT fibers show a Golgi signal with partial overlap of TGN38 with GM130. (**B**) KO fibers do not show an obvious overlap of the cis-Golgi protein GM130 with the trans-Golgi protein TGN38. (**C,D**) Representative images of skeletal muscle fibers from 5-month-old mice stained for GLG1 for (**C**) WT and (**D**) KO (scale bar: 10 µm). White dashed lines indicate cell borders. White arrows indicate co-localized structures. Right panel: schematic presentation of a skeletal muscle fiber with Golgi and Golgi-network positive structures. (**E**,**F**) GLG1 and TGN38 stainings in fibers reveal more trans-Golgi fragments in KO fibers with ImageJ analysis (n = 3 mice per group, 5 images per animal). (**E**) Particles were counted in the ImageJ menu “analyze particles” after converting pictures to a binary format. (**F**) Particles were counted via the ImageJ plugin ComDet v.0.5.5 without picture pre-processing. (**G,H**) The colocalization analysis reveals a partial overlap between GM130 and TGN38 positive particles in WT skeletal muscle fibers, which is reduced in KO samples (n = 3 mice per group, 5 images per animal). The colocalization analysis was performed either with the Coloc2 ImageJ plugin (**G**) that measures the Manderson overlap coefficient or the ComDet v.0.5.5 ImageJ plugin (**H**). 2-way-ANOVA with Bonferroni post-hoc analysis. (**I,J**) Representative images of Purkinje cells from 3-month-old mice stained for GM130 and TGN38 (scale bars: 5 µm) for (**I**) WT and (**J**) KO sections. White dashed lines indicate cell borders. White arrows indicate co-localized structures. (**K,L**) Representative images of (**K**) WT and (**L**) GMPPA KO MEFs stained for GM130 and TGN38 (scale bars: 5 µm). White arrows indicate co-localized structures. (**M**) ImageJ analysis (ComDet plugin) reveals increased numbers of TGN38 positive particles in Purkinje cells of KO mice (n = 4–5 mice per group, 10 images per animal). (**N**) The colocalization analysis (ComDet plugin) reveals a significantly reduced overlap between the cis-Golgi protein GM130 and the trans-Golgi protein TGN38 in Purkinje cells of KO mice (n = 4–5 mice per group, 10 images per animal, 2-way-ANOVA with Bonferroni post-hoc analysis). (**O,P**) ImageJ analysis (ComDet plugin) reveals an increased number of TGN38 positive particles (**O**) as well as a reduced colocalization between GM130 and TGN38 (**P**) in GMPPA KO MEFs (n = 4 experiments per group, 5 images per genotype per experiment). (**Q**) Representative electron microscopy images of skeletal muscle cells from 12-month-old WT and GMPPA KO mice (scale bars: 500 nm) showing the Golgi complex. Asterisks indicate abnormal secretory vesicles or Golgi cisterna. (**R**) Representative electron microscopy images of hippocampal neuronal cells from 12-month-old WT and GMPPA KO mice (scale bars: 500 nm) showing the Golgi complex. Asterisks indicate abnormal secretory vesicles or Golgi cisterna. (**S**) ImageJ analysis reveals reduced number of Golgi stacks and enlarged secretory vesicles in hippocampal neurons from KO mice (n = 3 mice per group, 4–10 images per animal, two-tailed Student’s t-test). * indicates *p* < 0.05, ** *p* < 0.01, and *** *p* < 0.001, **** *p* < 0.0001.

**Figure 2 biomedicines-11-00146-f002:**
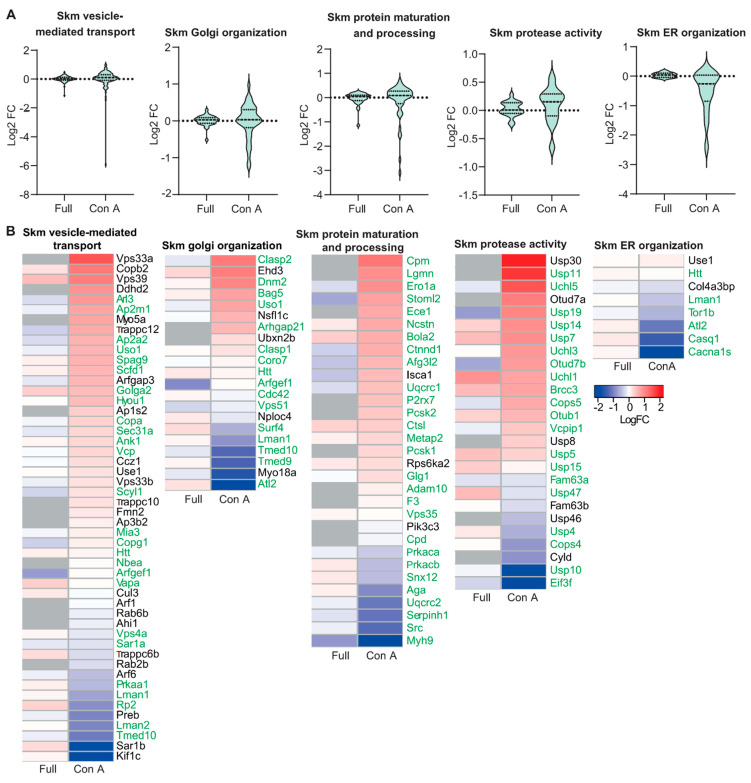
Quantitative changes of proteins necessary for ER and Golgi organization and function in GMPPA KO skeletal muscle compared to WT before and upon Con A pulldown. (**A**) Violin plots for proteins involved in selected biological processes (“Endoplasmic reticulum organization”, “Golgi organization”, “protease activity”, “protein maturation” and “protein processing”, and “vesicle mediated transport”) associated with the ER or the Golgi complex before and upon Con A enrichment (full: whole tissue lysates or Con A: Con A enriched samples). (**B**) Heatmaps for significantly altered proteins in either whole tissue lysates (full) or Con A enriched samples (Con A) (q-value below 0.05). Proteins significantly up-regulated between WT and GMPPA KO are shown in red and those down-regulated in blue (n = 3 mice per group). Not identified proteins are shown in grey. Green labeling indicates glycoproteins.

**Figure 3 biomedicines-11-00146-f003:**
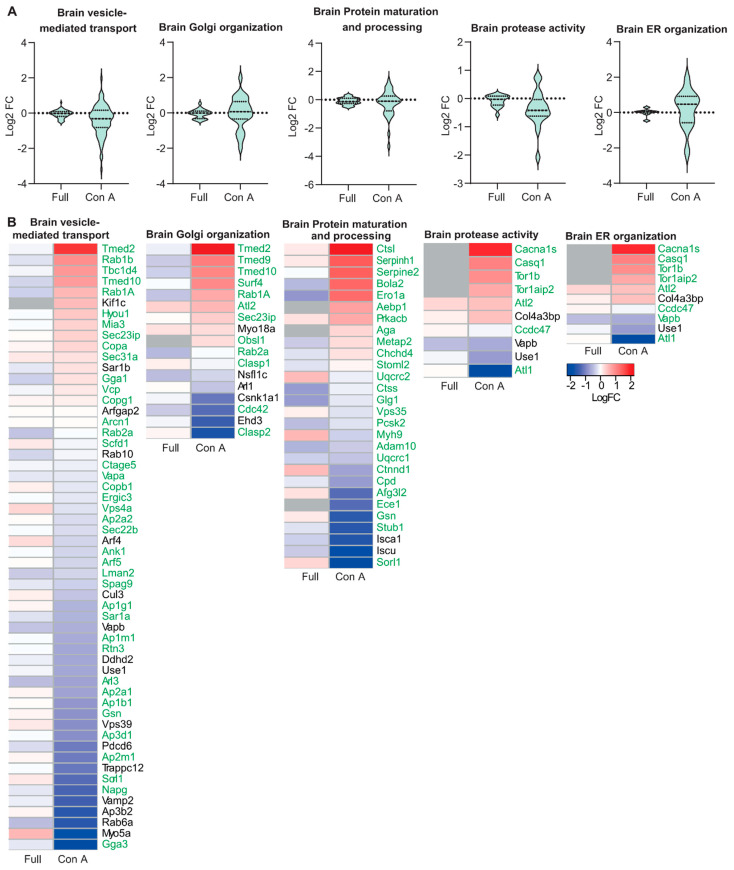
Quantitative changes of proteins necessary for ER and Golgi organization and function in GMPPA KO brain compared to WT before and upon Con A pull-down. (**A**) Violin plots for selected biological processes (“Endoplasmic reticulum organization”, “Golgi organization”, “protease activity”, “protein maturation” and “protein processing”, and “vesicle mediated transport”) associated with the ER or the Golgi complex before and upon Con A enrichment. (**B**) Heatmaps for significantly altered proteins found in either whole tissue lysates (full) or Con A enriched samples (Con A) (q-value below 0.05). Proteins significantly up-regulated between WT and GMPPA KO are shown in red and those down-regulated in blue (n = 3 mice per group). Not identified proteins are shown in grey. Green labeling indicates glycoproteins.

**Figure 4 biomedicines-11-00146-f004:**
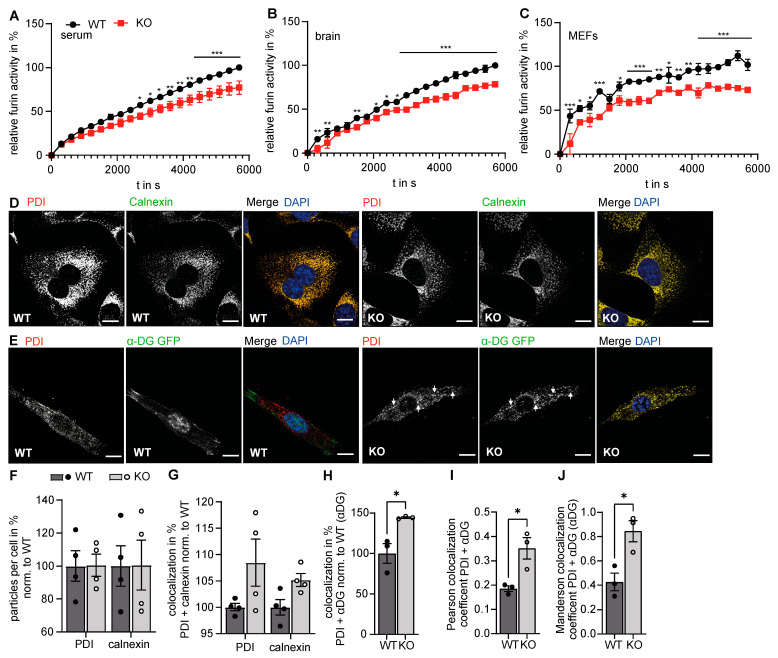
Loss of GMPPA results in functional alterations of the ER and the Golgi apparatus. (**A**–**C**) Furin activity assay for (**A**) serum (n = 4 mice per group), (**B**) brain lysates (n = 4 mice per group), and (**C**) MEF lysates (n = 3 experiments per group, 2-way-ANOVA with Bonferroni post-hoc analysis). (**D**) Representative images of WT and GMPPA KO MEFs stained for PDI and calnexin (scale bars: 5 µm). (**E**) Representative images of WT and GMPPA KO MEFs transfected with α-DG-GFP and stained for PDI (scale bars: 8 µm). White arrows indicate colocalized structures. (**F**,**G**) ImageJ analysis (ComDet plugin) does not show any differences between genotypes in (**F**) particle numbers as well as (**G**) colocalization between PDI and calnexin (n = 4 experiments per group, 5 images per genotype per experiment, 2-way-ANOVA with Bonferroni post-hoc analysis). (**H**) ImageJ analysis (ComDet plugin) shows increased colocalization between α-DG-GFP and PDI in KO cells (n = 3 experiments per group, 5 images per genotype per experiment, two tailed Student’s *t*-Test). (**I,J**) Colocalization analysis with the Coloc2 ImageJ plugin that measures the (I) Pearson’s colocalization coefficient as well as the (J) Manderson overlap coefficient showing increased colocalization between α-DG-GFP and PDI in KO cells (n = 3 experiments per group, 5 images per genotype per experiment, two tailed Student’s *t*-Test, and 2-way-ANOVA with Bonferroni post-hoc analysis). * indicates *p* < 0.05, ** *p* < 0.01, and *** *p* < 0.001.

**Figure 5 biomedicines-11-00146-f005:**
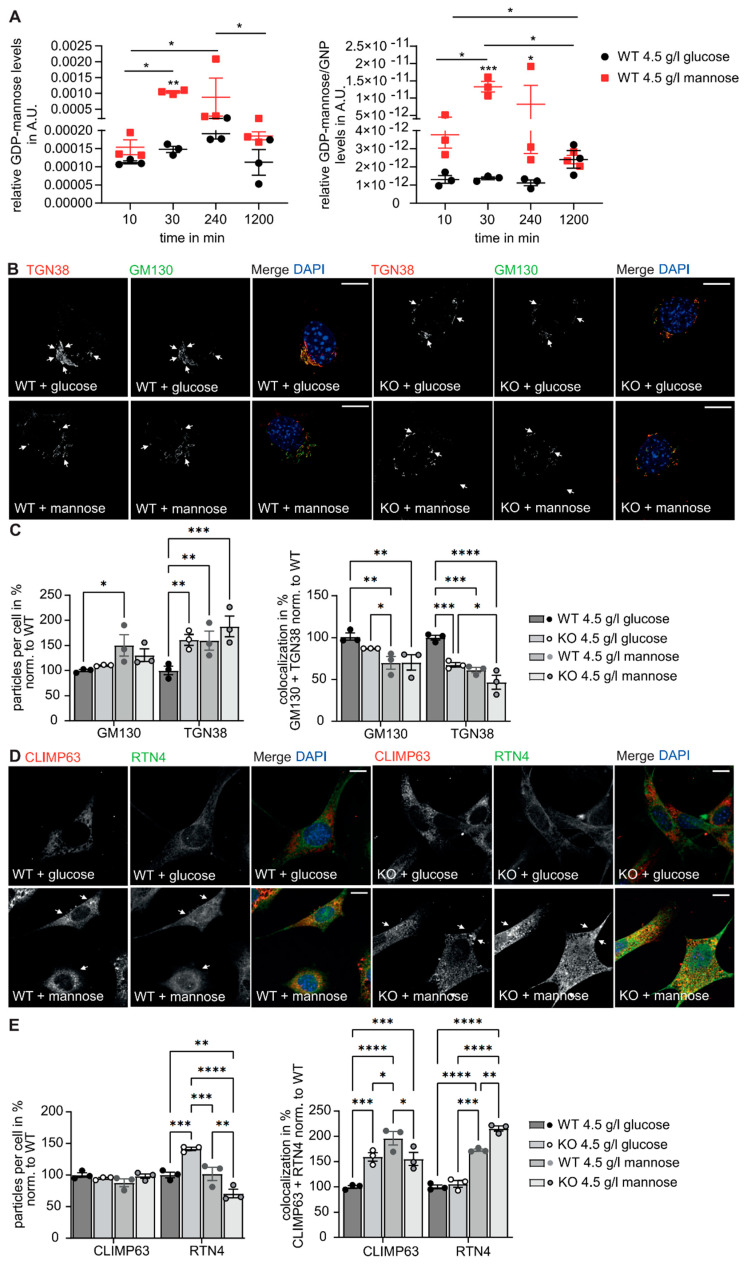
Effects of mannose supplementation on ER and Golgi morphology. (**A**) Left panel: relative GDP-mannose levels in arbitrary units (A.U.) measured in WT MEFs either cultured with 4.5 g/L glucose or mannose for different time points. Right panel: relative GDP-mannose levels in arbitrary units (A.U.) normalized to GNP levels (GMP + GDP + GTP) measured in WT MEFs either treated with 4.5 g/L glucose or mannose for different time points. (**B**) Representative images of WT and GMPPA KO MEFs stained for GM130 and TGN38 upon treatment with 4.5 g/L glucose or mannose (scale bars: 8 µm). White arrows indicate co-localized structures. (**C**) ImageJ analysis (ComDet plugin) for particle numbers and colocalization (n = 3 experiments per group, 9 images per genotype and treatment per experiment). (**D**) Representative images of WT and GMPPA KO MEFs stained for CLIMP63 and RTN4 upon treatment with either 4.5 g/L glucose or mannose (scale bars: 8 µm). White arrows indicate colocalized structures. (**E**) ImageJ analysis (ComDet plugin) for particle numbers and colocalization (n = 3 experiments per group, 2-way-ANOVA with Bonferroni post-hoc analysis). * indicates *p* < 0.05, ** *p* < 0.01, and *** *p* < 0.001, **** *p* < 0.0001.

**Figure 6 biomedicines-11-00146-f006:**
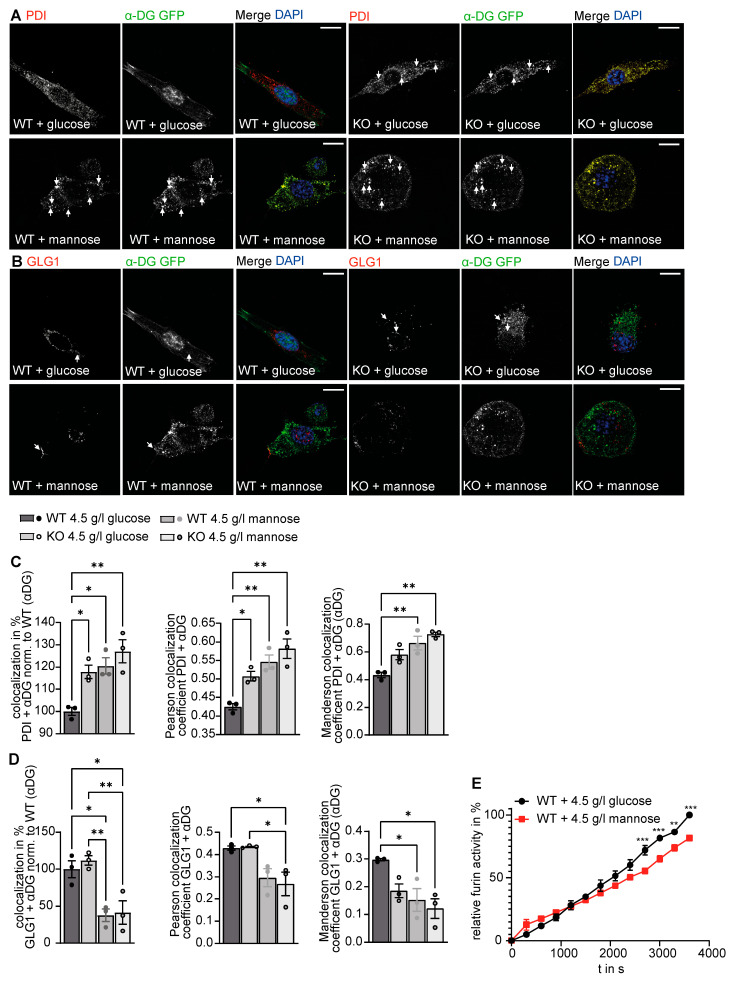
Culture of WT cells with mannose supplementation causes similar alterations of the secretory pathway as observed in GMPPA KO cells. (**A**) Representative images of WT and GMPPA KO MEFs transfected with α-DG-GFP and stained for PDI upon treatment with either 4.5 g/L glucose or mannose (scale bars: 8 µm). White arrows indicate colocalized structures. (**B**) Representative images of WT and GMPPA KO MEFs transfected with α-DG-GFP and stained for GLG1 cultured with either 4.5 g/L glucose or mannose (scale bars: 8 µm). White arrows indicate colocalized structures. (**C**) ImageJ analysis shows increased colocalization between α-DG-GFP and PDI in KO cells and upon mannose treatment (n = 3 experiments per group, 9 images per genotype and treatment per experiment). To analyze the colocalization between PDI and αDG, we either used the ComDet v.0.5.5 plugin (left), Pearson’s colocalization coefficient (middle), or the Manderson colocalization coefficient (right). (**D**) ImageJ analysis (ComDet plugin) shows decreased colocalization between α-DG-GFP and GLG1 upon mannose treatment (n = 3 experiments per group, 9 images per genotype and treatment per experiment). To analyze the colocalization between GLG1 and αDG, we either used the ComDet v.0.5.5 plugin (left), Pearson’s colocalization coefficient (middle), or the Manderson colocalization coefficient (right). 1-way ANOVA with Bonferroni post-hoc analysis. (**E**) Furin activity is reduced in WT MEFs upon replacement of glucose by mannose (n = 3 experiments per group, 2-way ANOVA with Bonferroni post-hoc analysis). * indicates *p* < 0.05, ** *p* < 0.01, and *** *p* < 0.001.

## Data Availability

All data are provided within the figures or in the Appendix A. Con A pull down mass spectrometry data are available at ProteomeXchange Consortium via the PRIDE (32) partner repository with the dataset identifier PXD014260 and PXD037129.

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
