# Peer review of "Impact of Hypermannosylation on the Structure and Functionality of the ER and the Golgi Complex"

_biomedicines, 2023, doi:10.3390/biomedicines11010146_

Round 1

Reviewer 1 Report

In this manuscript, the authors have investigated the architecture and function of the secretory pathway in cells derived from GDP-mannose-pyrophosphorylase A (GMPPA) k/o mice. These mice generated by the same group (Franzka et al, J Clin Invest 2021) recapitulates several features of the alacrima, achalasia, and mental retardation syndrome (AAMR). GMPPA is thought to act as an allosteric feedback inhibitor of GMPPB (GDP-mannose-pyrophosphorylase B) involved in the production of GDP-mannose, a core sugar required for glycosylation. As a consequence, loss of GMPPA results in an increase in the incorporation of

mannose residues into glycoproteins. The authors document that the morphology of the Golgi apparatus/TGN (Trans-Golgi Network) is altered in skeletal muscle fibers and in Purkinje cells derived from GMPPA k/o mice. Furin activity, that takes place in the TGN, is reduced in serum, brain and MEFs from GMPPA k/o mice, suggesting that Golgi function is affected beyond glycosylation. They also show that the alpha-dystroglycan (a-DG) transfected in MEFs seems to be more retained in the endoplasmic reticulum (ER) in GMPPA k/o than in WT cells. The addition of mannose in the culture medium of WT MEFs induces similar ER and Golgi alterations as observed GMPPA k/o cells. Finally, the authors performed a proteomic analysis after concanavalin A pull-down of ER and Golgi proteins in skeletal muscle and brain from GMPPA k/o and provides a list of proteins whose expression is modified.

My main concern is that this study is rather superficial and does not add much to our knowledge on the consequence of the hyper-mannosylation of proteins on the secretory pathway. In addition, some experiments do not support authors’ conclusions.

Specific comments:

1) Alterations of Golgi morphology is associated with many pathologies, including neurological diseases and metastasis. The interesting question is to understand why. In this study, it is only shown that furin activity is affected (and still in a moderate way). This could only be the consequence of the alteration of the morphology of TGN membranes and does not allow to conclude that “Golgi function is altered beyond glycosylation”. In some cases, a link (causality) has been established between alterations of the Golgi morphology and the expression of surface proteins/markers and/or altered signaling pathways. Similar experiments should be performed in wt and GMPPA-depleted cells.

2) In a previous work (Franzka et al, 2021), the authors showed that alpha-DG is hyperglycosylated in GMPPA k/o mice and that its turnover is increased in k/o myoblasts. In this study, they show that alpha-DG-GFP transfected in MEFs could be retained in the ER, as monitored by the co-localization with ER markers. However, fluorescence images were quantified using the Comdet plugin (Image J). This method is classically used to analyze colocalization of bright intensity spots. It may give reliable results in the case where alpha-DG-GFP co-localize with PDI in well-defined spots (Fig. 3E, k/o cells). In wild-type cells, alpha-DG-GFP staining is diffuse (presumably reflecting plasma membrane staining). I doubt that in that case the Comdet plugin gives accurate results. Other methods (Manders, Pearson,..) should be used. In the legend of Supp Fig. 3 (example of Comdet analysis), red/green/yellow circles are mentioned but they are not visible on the images. Another point is that, in k/o cells (Fig. 3E, white arrows), PDI colocalizes with alpha-DG-GFP in peripheral structures but these structures do not seem to be present in non-transfected cells (Fig. 3D). In addition, the morphology of transfected cells looks quite different than that of non-transfected WT or k/o cells. What is the reason for this? Altogether, this makes the experiments supporting the retention of alpha-DG-GFP in the ER not convincing. Other experiments (quantification of plasma membrane-associated alpha-DG-GFP, pulse-chase,…) should be performed.

- The authors provide a list of tens of proteins involved in many functions that are differentially expressed in WT and GMPPA k/o brain and skeletal muscle. This is likely not surprising. But the question is: How is this list informative and what does it tell us about the consequences of hyperglycosylation on ER/Golgi functions? In addition, a quick look at the list shows that there are proteins that are not, to my knowledge glycosylated, (Sar1a/b, Rab1a/b,…).

- It is not clear to me how the authors can conclude from the Supp Table I that “no changes in the overall protein abundance of ER- and Golgi-resident proteins was observed”.

Author Response

Dear reviewer,

thank you for your helpful feedback. Please see the attachment.

Reviewer 2 Report

This manuscript investigates the loss of mannose homeostasis in Golgi structure and function. Overall, this is a well-designed study with interesting data. The authors demonstrated an increase in mannosylated species in GMPPA KO mice along with a fragmented Golgi and decreased activity of TGN protease furin. Furthermore, WT cells supplemented with mannose phenocopies Golgi fragmentation which favors the authors finding that an imbalance in mannose homeostasis disrupts the Golgi integrity. For their experiments, the authors have used skeletal muscle and brain cells which seems appropriate as the GMPPA KO phenotypes are mostly skeletal and neurological.

Critique:

1.    ConA pull-down proteomics data revealed many up and down-regulated proteins, some of which are ER and Golgi-associated. Unfortunately, this data is not validated and not properly analyzed. Top “Golgi-associated” upregulated proteins (Spp1, Mink1, Cltb, Ocr1, Syap1, SNAP25, Cryab, Mmgt1) are not N-glycosylated and many of them are cytosolic. It is unclear how these proteins got recovered with ConA beads which suppose to bind mannose-containing N-glycosylated proteins. Without proper validation and analysis, this data is not convincing and should be removed from the paper.

2.    In GMPPA KOs there is a buildup of GDP-mannose while in the WT supplemented with, this may not be the case. The mechanism of free mannose toxicity could be different from GMPPA KO-induced toxicity because, in the latter, there is a build-up of the substrate that directly participates in glycosylation. Hence, it would be important for the authors to provide convincing data that a buildup of GDP-mannose results in an increase in mannosylated species in WT mannose-supplemented MEFs. Since they have used ConA in this study to immunoprecipitate mannosylated proteins, a ConA western blot should be done for this setup.

3.    To check the Golgi structure only the immunofluorescence confocal microscopy approach has been applied. It would be essential to supplement IF data with a proper electron microscopic approach.

4.    In the material methods 2.2 IF section, it would be important to indicate which objective lens has been used to take the confocal images.

5.    It has been mentioned that the colocalization between GM130 and TGN38 was decreased in skeletal muscle and Purkinje cells (Figure 1G, 1H, 1L, 1N). The proper description of the quantification has not been provided in the text or Figure legend.

6.    The cells were transiently transfected with alpha-DG in Figure 3E and it seems like alpha-DG is overexpressed in the cells. Overexpression may obscure the real findings.

Author Response

(The authors gave the same response as above.)

Reviewer 3 Report

Franzka and colleagues are exploring the effect of mannose on the secretory pathway. Previously they showed that GMPPA mutation causes the incorporation of mannose in different glycoproteins which leads to altered protein functions. In this manuscript, they checked the alteration in Golgi apparatus architecture and a few Golgi functions in GMPPA KO mice. Later they also checked the effect of Mannose supplementation on Golgi morphology. Overall, the data support the conclusions very well and the study is well-designed and executed, and I have only a few suggestions for improving the manuscript.

1. The Golgi staining is very poor, and the images are not well resolved. It’s very difficult to analyze for colocalization of the images when the author shows them at low magnification. Images need to be improved for publication.

2. The most appropriate experiment to test the Golgi morphology would be electron microscopy for WT and KO mice MEFs.

3. How do authors explain the increase in Golgi particle number in skeletal muscle fibers ? Is there any difference in Golgi particle size? How does the author explain the fragmentation of Golgi cisternae?

4. Figure 1M: How did the author quantify the particle numbers in MEFs with intact Golgi ribbon structure?

5. Is there any effect on secretion upon Golgi fragmentation due to mannose supplementation?

6. Fig 3E: Why there is a difference in PDI staining between Fig3D and E?

7. What does the author speculate on Golgi morphology in low or no mannose concentration conditions?

Author Response

(The authors gave the same response as above.)

Round 2

Reviewer 1 Report

The authors have met most of my previous comments in a satisfactory way.

Reviewer 2 Report

  1. The resolution of the IF images is better now. I was asking about the electron microscopic images and the author added the TEM images in Figure 1. Though the skeletal muscle TEM images of Golgi stacks look different. The Golgi seems swollen. Otherwise it is a good addition.
  2. The authors have reanalyzed the MS data. They have concluded that the mentioned proteins are not glycosylated but interact with glycoprotein. They have generated violin plots for whole tissue lysates versus Con A enriched samples for better visualization of protein alterations (Figure 2A and 3A). I think this approach is a better addition to MS analysis. I think they did not want to delete the MS dataset as they had already performed the experiment. In support, they reanalyzed MS analysis which seems fine to me.
  3. The author explained the Golgi particle analysis now which was missing before.
  4. To show a buildup of GDP-mannose results in an increase in mannosylated species in WT mannose-supplemented MEFs the authors have added WB data probed for ConA in a supplementary figure which is great.
  5. They have done a colocalization analysis in the revised manuscript which seems adequate.
  6. The transient transfection of MEF cells with alpha-DG can obscure real findings,      but the authors mentioned the limitation regarding endogenous expression.

Overall I found the revised manuscript better than the previous one.

Reviewer 3 Report

The authors responded to all my comments and the manuscript can be accepted in its present form.